# Assessing the Livelihood Vulnerability of Nomads to Changing Climate in the Third Pole Region of Nepal

Rijan Bhakta Kayastha [1], Woo-Kyun Lee [2], Nischal Shrestha [1] and Sonam Wangyel Wang [3,*]

1    Himalayan Cryosphere, Climate and Disaster Research Center (HiCCDRC), Department of Environmental Science and Engineering, School of Science, Kathmandu University, Dhulikhel 45200, Nepal; rijan@ku.edu.np (R.B.K.); n.shrestha@bioconnepal.org (N.S.)
2    Division of Environmental Science and Ecological Engineering, Seoul 02841, Republic of Korea; leewk@korea.ac.kr
3    OJeong Resilience Institute and Key Research Institute (Project Third Pole), Seoul 02841, Republic of Korea
*    Correspondence: wangsonam@korea.ac.kr

**Abstract:** This study was performed in Nepal's Langtang and Shey Phoksundo National Parks (NPs) to find out how vulnerable the nomads' ways of making a living are to climate change. We interviewed 68 household heads between March and May 2022 to obtain information on 13 components and 46 indicators. The original data were backed up by an analysis of the published and unpublished literature that was available. A composite index was used to combine the data, and different vulnerabilities were compared. As nomads in both NPs rely mostly on natural resources for energy, water, and food, the findings revealed that land, energy, water, sanitation, and natural resources are the most important factors influencing nomads' livelihood vulnerability in both NPs. Although herders in Shey Phoksundo NP suffered less loss as a result of climate change-related natural disasters, human–wildlife conflict was a major issue in both parks. Both the livelihood vulnerability index (LVI) and LVI–IPCC suggested that both national parks were moderately vulnerable to climate change indicators. The results are likely to serve as empirical evidence for future strategies, such as implementing policy measures aimed at reducing the sensitivity of habitat conditions, increasing societal resilience, introducing sustainable livelihood alternatives, and improving individual stability.

**Keywords:** herders; Himalayas; climate change; livelihood vulnerability index; sensitivity; exposure; adaptive capacity





## 1. Introduction

Global climate change is becoming the biggest threat to the health of people and the environment on Earth [1]. It is expected to make natural disasters happen more often and be worse. It will also bring new dangers, such as rising sea levels, melting glaciers, and drying water sources [2]. Current climatic shocks and pressures have already had a significant influence on household vulnerability, particularly in rural areas [3–5]. Local indigenous communities are strategically positioned in the Himalayan foothills' plains and highlands, where indigenous knowledge and adaptation are most advantageous. Unfortunately, climate change hazards, such as expanding mountainous threats, water scarcity caused by glacier melting, and avalanches, are more prevalent in these areas and could destroy communities, key infrastructure, ecosystem services, livelihoods, health, and other aspects of human well-being [1,6]. As a result of their greater exposure to natural disasters, residents of these areas have a higher risk of casualties. Despite these challenges, people have become accustomed to living in these places because of cultural interaction, tourism, and job creation. Therefore, growth is unavoidable. This entails addressing specific adaptations as well as reducing challenges and opportunities and mainstreaming them into long-term development programs. Despite the urgency, scientific research that provides a

foundation for livelihood trajectories, as well as different adaptive options and capacities, is scarce [7].

The Pamir, Kunlun, Tian Shan, and Altai mountains, which make up Asia's High Mountains (AHM), are the headwaters of river systems that give fresh water to one-third of the world's population [8]. These mountains are ecologically significant because they provide a primary habitat and migration route for the rich biodiversity, including the endangered snow leopard (*Panthera uncia*), that the landscape is endowed with [9]. This region is an inevitable habitat for wildlife and mankind, with significant global benefits that must not be jeopardized. The Hindu Kush Himalayas (HKH) are one important mountain range that comprises the AHM. Hundreds of biodiversity hotspots mark the HKH region, providing critical ecological services both directly to inhabitants and indirectly to the global population. The vast majority of people are agriculturalists who rely on natural resources for drinking water, food, electricity, and other ecosystem services such as spirituality [10]. This tight connectivity, which leads to the over-exploitation of natural resources and vulnerability to extreme weather events, is fast eroding both ecological and social systems' inherent capacities, rendering them less resilient to disasters. Global warming and climate change worsen this vulnerability even further in these fragile HKH regions [6]. These regions are likely to endure at least three times the amount of warmth as the global climate continues to increase [11]. Indigenous populations in the Himalayas that are heavily reliant on natural resources are among the first and most impacted by climate change [12]. However, these mountainous areas of the Himalayan region have received little attention in terms of how climate change and related risks affect the livelihoods of marginalized communities [5]. Warming is higher in the mountains compared to anywhere on earth [13], thereby impacting the nomads, livestock, biodiversity, and the environment severely [14]. In the face of a rapidly changing climate, the melting of the region's huge glacier fields, along with unpredictable rainfall, is affecting river flows and seasonal water availability [15–17]. As a result, endangered species, local and downstream people, as well as agricultural output all suffer. Greater strain on high mountain ecosystems is caused by poor water resource management, land degradation, forest and grassland fragmentation and loss, wildlife hunting, and livestock overgrazing, all of which lead to increased human–wildlife conflict [18]. As a result, inhabitants living at high elevations in this region have a genuine interest in decreasing their susceptibility to climate change and the numerous threats to Asia's high mountain ecosystems [19].

There is an urgent need to put resources into better understanding the region's ecological and social systems using proven scientific methods so that science-based adaptation interventions can be used to build sustainable ecological and social systems so that future generations can continue to enjoy local and global environmental benefits. This quest needs a thorough examination of vulnerabilities and disaster risks in the HKH area of the Third Pole. In many contexts, vulnerability assessments have become a key tool for assessing development challenges and the impact of climate change. Such assessments could include a variety of methods for systematically considering interactions between people and their environments, including both physical and social factors [20]. The Langtang NP and Shey Phoksundo NP in Nepal have been home to indigenous communities for centuries, living in high-elevation settlements that make them vulnerable to the impacts of modernization and globalization. Many have abandoned traditional agriculture and livelihoods in favor of tourism or seeking employment abroad, leaving those who continue to rely on traditional subsistence methods even more vulnerable. This unique setting offers an ideal context to examine the vulnerability of the last remaining nomads. We used the livelihood vulnerability index (LVI) and LVI–IPCC to assess the livelihood vulnerability of herders living in isolated, high-elevation settlements in two national parks in the central Himalayan region of Nepal. We chose these tools because of their efficacy in evaluating and managing the impact of climate change on marginalized communities [4]. Using a variety of different factors, this study looked at how gender, job, and other livelihood aspects are affected by climate change and how sensitive a household is to climate change.

## 2. Materials and Methods

### 2.1. Study Area

The study was conducted in Langtang National Park (Lantang NP) and Shey Phoksundo National Park (Shey Phoksundo NP), located in the high mountains of Nepal (Figure 1). Langtang NP was established in 1976 and covers three districts: Rasuwa, Nuwakot, and Sindhupalchowk, with an area of 1710 km$^2$ and a buffer zone of 420 km$^2$. The park's ecosystems range from subtropical forest to alpine scrub and perennial ice, with an altitudinal range of 800 to 7245 m and a remarkable variety of 14 vegetation types in 18 ecosystem types that are home to rare and endangered wildlife, such as red pandas and snow leopards [21,22]. The Langtang region is well known for its pristine forests, high-altitude pastures of wild sheep, and breathtaking mountain views. Animal husbandry has been the main source of income for the Tibetan-speaking community in Langtang for at least 300 years [23].

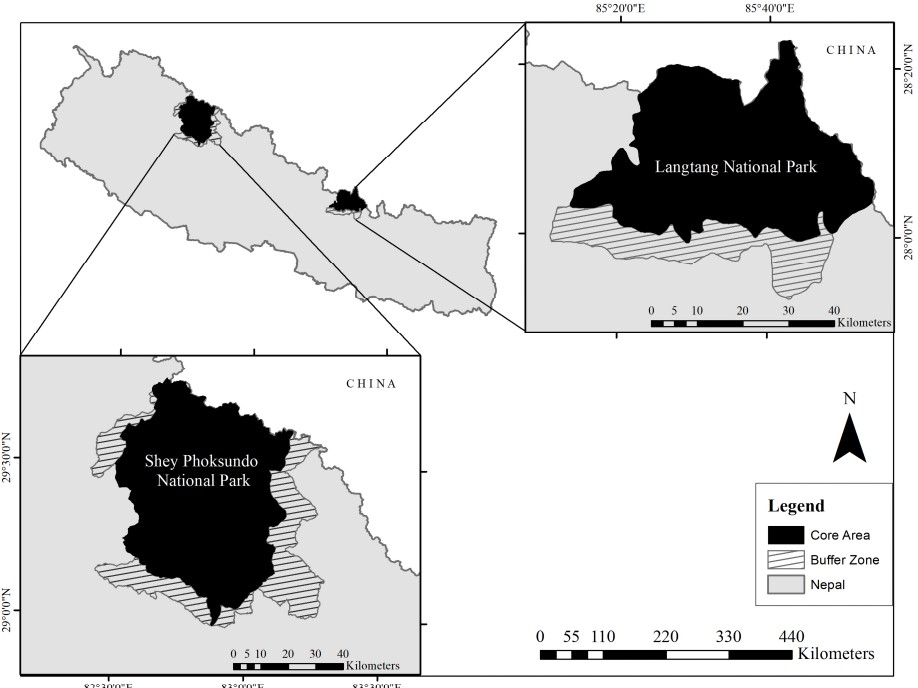

**Figure 1.** Study area map showing Langtang and Shey Phoksundo NPs in Nepal.

Shey Phoksundo NP was established in 1984 and is the largest national park in the midwestern part of Nepal. It is 3555 km$^2$ in size and has a buffer zone of 1349 km$^2$ around it. The national park encompasses Nepal's Dolpa and Mugu districts [24]. One-third of the park territory is the physiography of steep, jagged Himalayan mountains, reaching a height of 6883 m at Mount Kanjiroba. The majority of Shey Phoksundo National Park's terrain is composed of high, gently sloping hills that merge into the Tibetan Plateau and steppe, with abundant rhododendron, caragana bushes, and salix on the slope areas, which are home to some endangered species such as snow leopards, grey wolves, and musk deer [21,25]. Shey Phoksundo NP also has people that live a transhumant pastoral lifestyle, with the local economy built on agro-pastoralism and cattle husbandry as the major sources of income, food, and transportation [26]. In this context, interviews were carried out with traditional herders in Langtang NP and Shey Phoksundo NP, respectively.

### 2.2. Sampling and Data Collection

The study was based on primary data collected from March 20 to 28, 2022 (for Langtang NP-17 households) and May 22 to 29, 2022 (for Shey Phoksundo NP-51 households) using a pre-tested, open-ended, semi-structured, and in-depth interview to obtain qualitative information about the dynamics of livelihood vulnerability scenarios. After receiving

authorization from the respective national park offices, trained field staff conducted interviews in the Nepalese language with the various household heads. Before conducting the survey, the purpose of the study and verbal consent to conduct the survey were obtained from the head of the household. Unwilling respondents were not questioned. Community leaders were contacted for this purpose, as well as to translate the questions into their local language when possible. Snowball sampling procedures were used to select households in both national parks. A detailed list of the major and sub-components used in the survey can be seen in Table 1. For the climate data analysis, the daily minimum and maximum temperatures at Dhunche station from 1989 to 2017 and rainfall data from 1980 to 2017 were used in the case of Langtang NP. Similarly, daily minimum and maximum temperatures and rainfall data from Dunai station from 1985 to 2017 are used in Shey Phoksundo NP.

**Table 1.** The contribution of the LVI thirteen major components to the LVI–IPCC.

| LVI Major Components | IPCC Definition of Vulnerability (LVI–IPCC) |
|---|---|
| Climate change and disasters<br>Human–wildlife Conflict | Exposure |
| Socio-demographic profile<br>Natural resources<br>Energy<br>Infrastructure<br>Land<br>Social networks<br>Finance and income | Adaptive Capacity |
| Health<br>Water and sanitation<br>Agriculture and food security<br>Housing | Sensitivity |

*2.3. Data Analysis*

2.3.1. Calculation of Livelihood Vulnerability Index (LVI)

The LVI is divided into thirteen sections: socio-demographic profile (SDP), agriculture and food security (AFS), land (L), infrastructure (I), housing (HO), natural resources (NR), energy (E), social networks (SN), health (H), finance and income (FI), water and sanitation (WS), human–wildlife conflict (HWC), or climate change and disasters (CCD). Each one is made up of several indicators or sub-components. These sub-components can provide a more comprehensive assessment of vulnerability and facilitate the development of targeted interventions to address specific vulnerabilities. These were created based on an assessment of the literature on each key component as well as the feasibility of collecting the data needed through household surveys. The LVI adopts a balanced weighted average technique [27], in which each sub-component contributes equally to the overall index, even though each main component includes a varying number of sub-components.

Each LVI requires four steps to calculate. The first step was to convert the raw data into useful measurement units such as percentages, ratios, and indices. Because the sub-components are measured on different scales, step 2 was to standardize them. To aggregate all measurements into a single LVI index, this was essential. The calculation for this conversion was modified from the one used to generate the Human Development Index's life expectancy index. The quotient of the difference between the actual score and the minimum value received from the whole sample, as well as the difference between the maximum and minimum values acquired from the complete sample, were computed to normalize a key component as shown in Equation (1):

$$index_{S_d} = \frac{S_d - S_{min}}{S_{max} - S_{min}} \tag{1}$$

where $S_d$ is the original sub-component and $S_{min}$ and $S_{max}$ are the minimum and maximum values for each sub-component, respectively.

In Step 3, the average of each main component's standardized scores was determined, yielding a final score for each key component (Equation (2)).

$$M_d = \frac{\sum_{i=1}^{n} index_{sdi}}{n} \tag{2}$$

where $M_d$ is one of thirteen major components for national parks (socio-demographic profile (SDP), agriculture and food security (AFS), land (L), infrastructure (I), housing (HO), natural resources (NR), energy (E), social networks (SN), health (H), finance and income (FI), water and sanitation (WS), Human–wildlife conflict (HWC), or climate change and disasters (CCD)), $index_{sdi}$ is the number of sub-components. Step 4 generated the LVI score by combining the weighted averages of all the key components. The weights of each primary component were calculated based on the amount of indication it contained (Equation (3)).

$$LVI_d = \frac{\sum_{i=1}^{13} w_{Mi} M_{di}}{\sum_{i=1}^{13} W_{Mi}} \tag{3}$$

where $LVI_d$, or the livelihood vulnerability index for national parks d, is the weighted average of the eight primary components. The weights of each major component, $W_{Mi}$, are determined by the number of sub-components that comprise each major component and are included to guarantee that all sub-components contribute equally to the total LVI [27]. In this study, the LVI is scaled from 0 (least vulnerable) to 1 (most vulnerable). Equation (4) can be used to express this as follows:

$$LVI_d = \frac{w_{SDP}SDP_d + w_{AFS}AFS_d + w_L L_d + w_I I_d + w_{HO}HO_d + w_{NR}NR_d + w_E E_d + w_{SN}SN_d + w_H H_d + w_{FI}FI_d + w_{WS}WS_d + w_{HWC}HWC_d + w_{CCD}CCD_d}{w_{SDP} + w_{AFS} + w_L + w_I + w_{HO} + w_{NR} + w_E + w_{SN} + w_H + w_{FI} + w_{WS} + w_{HWC} + w_{CCD}} \tag{4}$$

### 2.3.2. Calculating LVI based on IPCC Framework (LVI–IPCC)

The livelihood vulnerability index–Intergovernmental Panel on Climate Change (IPCC) is an alternative approach for calculating LVI according to the IPCC definition of vulnerability [28]. We devised a new approach for computing the LVI that takes into account the IPCC vulnerability criteria. The eight fundamental components of the LVI–IPCC framework are organized in Table 1. The index of exposure (Exp) contains climate change and disasters along with Human–wildlife conflict; sensitivity (Sen) contains health, housing, agriculture and food security, and water and sanitation; and adaptive capacity (Adp. Cap) contains the socio-demographic profile, social network, and finance and income. LVI–IPCC is calculated differently than the primary components of LVI combined. To begin, all components in Table 1 will be combined by category plan using Equation (5):

$$CF_d = \frac{\sum_{i=1}^{n} W_{Mi} M_{di}}{\sum_{i=1}^{n} W_{Mi}} \tag{5}$$

where $CF_d$ represents the contributing factors according to IPCC (exposure, sensitivity, or adaptive capacity) for area d; $M_{di}$ represents a main component for area *d*, which is indexed by I; $W_{Mi}$ represents the quality of the main component; and *n* represents the number of the main components of each contributing factor. The combination of these three contributing factors is calculated using Equation (6):

$$LVI - IPCC_d = (exposure_d - adaptive\ capacity_d) * sensitivity_d \tag{6}$$

where $LVI$-$IPPC_d$ represents the *LVI* index in area d expressed by using the framework of the vulnerability of the IPCC. The scale of LVI–IPCC ranges between (−1) and (−0.4) is not vulnerable; (−0.41) and (0.3) are vulnerable or moderate, and (0.31) and (1) are very vulnerable.

## 3. Results

### 3.1. Livelihood Vulnerability Index (LVI)

Overall, we found natural resources, energy, water and sanitation, and land to be the most important components affecting the livelihood vulnerability of the NPs in Nepal, as shown in Table 2. Results show that the national parks had some similarities and some variances, as predicted. Respondents from both national parks were primarily reliant on their own farming for food; social network indicators were similar, and most respondents had access to information via television, radio, cell phone, or the internet. Both national parks' respondents were heavily reliant on agriculture as their primary source of income, and they relied entirely on natural resources for energy. Table 2 and Figure 2 show the major components comprising the composite LVI of the Langtang and Shey Phoksundo NPs. The overall LVIs of the Langtang and Shey Phoksundo NPs are very similar at 0.53 and 0.52, respectively.

**Table 2.** Normalized major components, sub-components, and overall LVI for nomads of Langtang and Shey Phoksundo NPs in Nepal.

| Major Components | Sub-Components | Units | Langtang NP (*n* = 17) | Shey Phoksundo NP (*n* = 51) |
|---|---|---|---|---|
| Socio-Demographic | | | 0.46 | 0.36 |
| | Dependency Ratio | Ratio | 0.538 | 0.22 |
| | % of HH heads that did not attend school | Percent | 100 | 77 |
| | % of female-headed HHs | Percent | 29.41 | 22.64 |
| | Average age of HH heads | Years | 47.76 | 50.49 |
| | Average age of female HH heads | Years | 48.8 | 44.41 |
| | Average HH size | Number | 6.35 | 5.28 |
| | % of HHs with family members working outside the community | Percent | 11.76 | 13.21 |
| Social Networks | | | 0.69 | 0.51 |
| | % of HH members being a part of any community-based groups | Percent | 23.53 | 0 |
| | % of HHs owning a mobile phone | Percent | 76.47 | 100 |
| | % of HHs without radios | Percent | 94.12 | 0 |
| | % of HHs that have not received local government assistance in the past 12 month | Percent | 82.35 | 100 |
| Health | | | 0.34 | 0.09 |
| | Average time to reach the nearest health center | Minutes | 56.17 | 138.11 |
| | % of HHs having a chronically sick family member (they get sick often) | Percent | 23.53 | 0 |
| Agriculture and Food Security | | | 0.42 | 0.39 |
| | % of HHs where agriculture and livestock grazing is the only source of income | Percent | 52.94 | 24.53 |
| | % of HHs that are food sufficient | Percent | 100 | 100 |
| | Average livestock units owned by HHs | Number | 23.41 | 17.62 |
| | Average agricultural land units owned by HHs | Acres | 1.65 | 1.65 |
| | Average agricultural livelihood diversification index | Number | 0.33 | 0.33 |

**Table 2.** *Cont.*

| Major Components | Sub-Components | Units | Langtang NP (*n* = 17) | Shey Phoksundo NP (*n* = 51) |
|---|---|---|---|---|
| Land | | | 0.72 | 0.63 |
| | % of HHs owning pastureland | Percent | 29.41 | 0 |
| | % of HHs having pasture shortage | Percent | 70.58 | 100 |
| | % of HHs observing pastureland degradation in the last 10 years | Percent | 100 | 100 |
| | % of HHs believing weather/climate change is the reason for pasture degradation | Percent | 88.23 | 52.83 |
| Infrastructure | | | 0.28 | 0.73 |
| | Average time to reach the nearest vehicle station or bus stop | Minutes | 352.94 | 453.2 |
| | Average time to reach the nearest market | Minutes | 352.94 | 513.96 |
| Housing Type | | | 0.28 | 0.0 |
| | % of HHs living in temporary housing | Percent | 23.53 | 0 |
| Natural Resources | | | 1.0 | 1.0 |
| | % of HHs depending on natural resources | Percent | 100 | 100 |
| Energy | | | 1.0 | 1.0 |
| | % of HHs using forest-based fuel for cooking and heating | Percent | 100 | 100 |
| | % of HHs with sufficient fuel | Percent | 100 | 100 |
| Finance and Income | | | 0.38 | 0.74 |
| | % of HHs owing money to anyone in the last 1 year | Percent | 11.76 | 47.17 |
| | % of HHs having access to loan | Percent | 64.70 | 100 |
| Water and Sanitation | | | 0.70 | 0.75 |
| | % of HHs having access to clean and safe drinking water (natural source/jungle) | Percent | 100 | 100 |
| | % of HHs reporting water conflicts within their local community | Percent | 11.76 | 0 |
| | % of HHs having enough drinking water for livestock | Percent | 94.12 | 100 |
| | % of HHs having access to private toilets | Percent | 76.47 | 100 |
| Climate Change and Disasters | | | 0.31 | 0.25 |
| | % of HHs observing or experiencing natural hazards in the last 5 years | Percent | 70.58 | 13.21 |
| | % of HHs receiving advanced climate warning | Percent | 0 | 0 |
| | % of HHs suffering any loss (agriculture or livestock) to natural disasters, including death, in the last five years | Percent | 17.65 | 50 |
| | % of HHs receiving compensation from related agencies on the loss and damages of assets due to climate change disasters | Percent | 5.88 | 0 |
| | % of HHs losing livestock/crop or human life to climate and weather-related events only | Percent | 41.17 | 3.77 |
| | Average number of livestock lost due to climate and weather-related events | Number | 0 | 1.11 |
| | Mean standard deviation of monthly average maximum temperature | °C | 2.29 | 4.18 |
| | Mean standard deviation of monthly average minimum temperature | °C | 2.04 | 3.68 |
| | Mean standard deviation of monthly precipitation | mm | 8.11 | 2.33 |

**Table 2.** *Cont.*

| Major Components | Sub-Components | Units | Langtang NP (*n* = 17) | Shey Phoksundo NP (*n* = 51) |
|---|---|---|---|---|
| Human–wildlife Conflict | | | 0.33 | 0.31 |
| | % of HHs losing livestock or human life to wildlife | Percent | 47.06 | 58.49 |
| | Average number of livestock predated by wildlife | Number | 1.53 | 1.75 |
| | % of HHs whose agricultural fields/crops were damaged by wildlife | Percent | 47.06 | 13.21 |

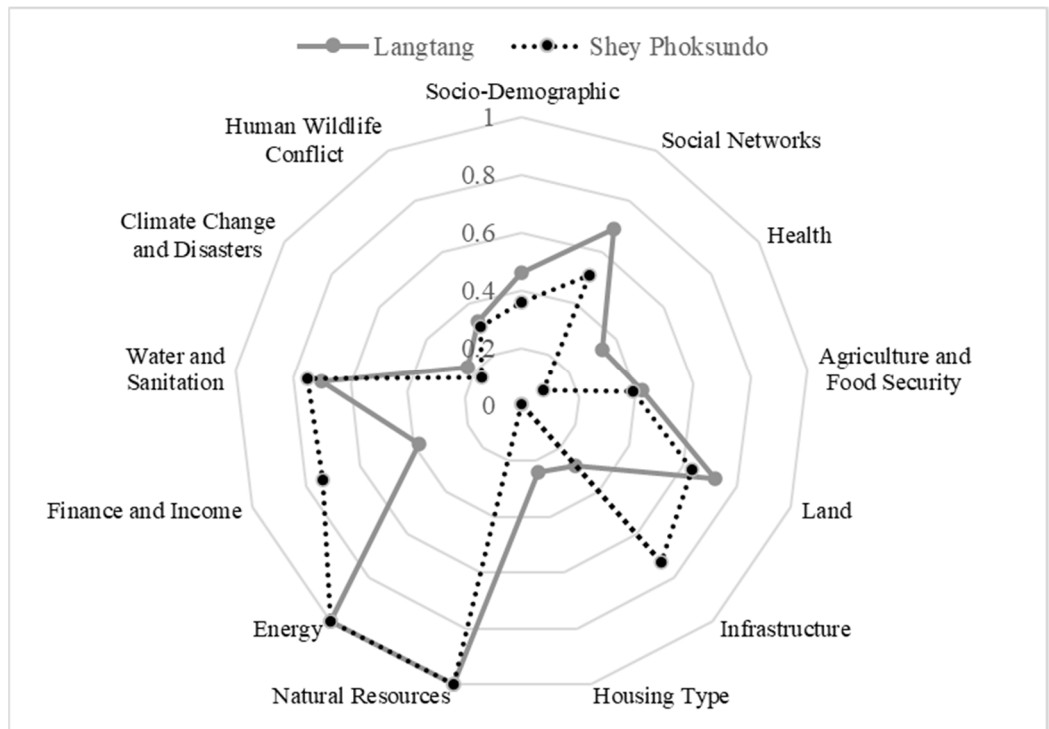

**Figure 2.** A spider diagram showing various degrees of importance (on a 0 to 1 scale) of thirteen components of LVI in Langtang and Shey Phoksundo NPs in Nepal.

The dependency ratio of Langtang is significantly higher (0.538) compared to Shey Phoksundo NP (0.22), which shows that children and people over 65 are more numerous in Langtang NP. In the Langtang NP survey, all the household heads were illiterate, but just one-third of the household heads in Shey Phoksundo NP had some schooling. Both the national parks had very few female household heads, with an average age of 48.8 and 44.41 in Langtang NP and Shey Phoksundo NP, respectively. In Langtang NP, this was greater than the average household head age, whereas, in Shey Phoksundo NP, it was lower. In both NPs, the typical family size was between 5 and 6, and practically every family member worked in the community.

Compared to Shey Phoksundo NP (0.51), Langtang NP (0.69) exhibited a higher LVI for social networks. Nearly one-fourth of the houses in Langtang NP had their family members as a part of any community-based groups, whereas not a single member was associated with community-based groups in Shey Phoksundo NP. Every household in Shey Phoksundo NP had a mobile phone, but none of them had a radio. On the other hand, only one-fourth of households in Langtang NP did not have mobile phones, although almost every household had a radio. In Langtang NP, just a few households received any sort of assistance from the local government in the previous 12 months, but none of the households in Shey Phoksundo NP did.

The composite Health LVI of Langtang (0.34) is more than three times that of Shey Phoksundo (0.09). Nomads in Langtang NP took roughly an hour on average for individuals to reach the nearest health clinic, whereas it took more than twice as long in Shey Phoksundo NP. In Langtang NP, one in four homes had a family member who was always sick, but this did not happen in Shey Phoksundo NP.

Only approximately a quarter of the households in Shey Phoksundo NP relied only on agriculture and livestock grazing for income, whereas over half of the households in Langtang NP relied primarily on agriculture and livestock grazing for income. In both national parks, however, all the families were food self-sufficient. Households in both national parks owned an average of 17–24 livestock and 1.65 acres of land. Each national park had an average agricultural livelihood diversification index of 0.33 because there were only two ways to make a living from farming in each park. In Langtang NP, over a third of families held pasture lands, whereas not a single household in Shey Phoksundo NP did. Every household in Shey Phoksundo NP was affected by pasture scarcity, although just 30% of households in Langtang NP were affected. All of the households in both NPs have seen the degradation of pastureland during the last ten years. Additionally, only half of the families in Shey Phoksundo NP believe that weather or climate change is to blame for the deterioration, but over 90% of the households in Langtang NP agree.

On average, household members in Langtang NP took approximately 6 h to reach the nearest market and vehicle stop, but household members in Shey Phoksundo NP took longer in both aspects. It took them roughly 7 h and 30 min to reach the nearest vehicle station, and an average of 8 h and 30 min to get to the nearest market. In Shey Phoksundo NP, not a single household was living in temporary shelters, but there were a quarter of households living in such shelters in the case of Langtang NP. Every household in both NPs relied entirely on natural resources for food, energy, and income. Every home in both national parks used wood to cook and heat their homes, and there was plenty of it. All households in both national parks have access to clean and safe drinking water from natural springs. They also had enough water for their livestock. Except for a few occurrences in Langtang NP, there were no water-related disputes in the national parks. In Langtang NP, one-quarter of the families had access to toilets, whereas every household in Shey Phoksundo has a good toilet facility. Only about 10% of households in Langtang NP owed money to anybody in the previous year, but the situation was different in Shey Phoksundo NP, where about 50% of households owed money to anyone. Everyone in Shey Phoksundo NP had access to loans, but the same was true for only around 60% in Langtang NP.

In the past five years, over two-thirds of households in Langtang NP witnessed or experienced natural hazards, but barely more than 10% of households in Shey Phoksundo NP did. This resulted in agriculture or livestock losses for half of the families in Shey Phoksundo NP and approximately 17% of the households in Langtang NP. Similarly, almost 40% of households in Langtang NP lost livestock/crop or human life due to climatic and weather-related events alone, while Shey Phoksundo NP only lost 3%. However, none of the households received an advanced climate warning. Almost 6% of households in Langtang NP received compensation from related agencies for asset losses and damage caused by climate change disasters, but no such cases were observed in Shey Phoksundo NP. More than 50% and almost 60% of households in the Langtang and Shey Phoksundo National Parks, respectively, reported livestock or human life loss owing to wildlife, especially snow leopards (*n* = 33). In each NP, animals preyed on an average of fewer than two livestock per day. In Langtang National Park, approximately 60% of households reported wildlife damage to their agricultural fields and crops by monkeys (*n* = 9), whereas the incidence was significantly lower in Shey Phoksundo National Park. Only slightly more than 10% of households in the latter park reported crop loss due to wildlife, particularly monkeys (*n* = 6).

### 3.2. LVI–IPCC

The overall LVI based on the IPCC framework of Langtang NP and Shey Phoksundo NP is −0.145 and −0.133, respectively, which denotes that both national parks are moderately vulnerable, as shown in Table 3. Figure 3 shows a vulnerability triangle, which plots the contributing factor scores for exposure, adaptive capacity, and sensitivity. The triangle indicates that Langtang NP is more exposed (0.32) to the impacts of climate change than Shey Phoksundo NP (0.28). Similarly, Langtang NP (0.44) is more sensitive to climate change impacts than Shey Phoksundo NP (0.31). Furthermore, Langtang NP (0.65) has a lower adaptive capacity to the impacts of climate change than Shey Phoksundo NP (0.71). In such an analysis, the index results should be viewed as relative values that can only be compared within the study sample.

**Table 3.** LVI–IPCC contributing factors for calculating for studied national parks in Nepal.

| IPCC Definition of Vulnerability (LVI–IPCC) | Langtang NP | Shey Phoksundo NP |
| --- | --- | --- |
| Exposure | 0.32 | 0.28 |
| Adaptive Capacity | 0.65 | 0.71 |
| Sensitivity | 0.44 | 0.31 |
| LVI–IPCC | −0.145 | −0.133 |

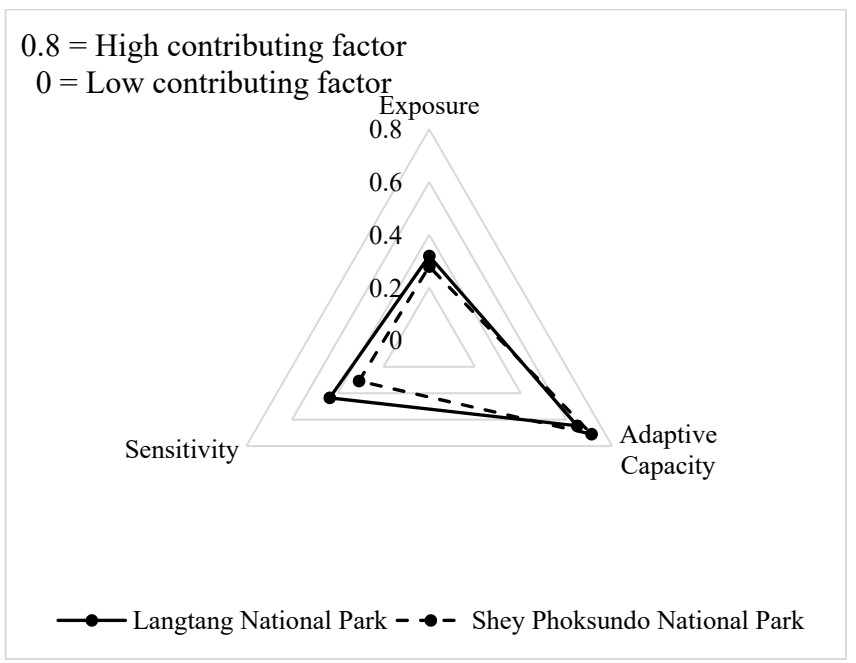

**Figure 3.** Vulnerability triangle diagram of the contributing factors of the livelihood vulnerability index–IPCC (LVI–IPCC) for Langtang and Shey Phoksundo NPs, Nepal.

## 4. Discussion

We used primary data from household surveys in both national parks to assess how vulnerable the nomads and their ways of making a living are to environmental changes, including climate change. While LVI–IPCC identified the studied community's adaptive capacity, sensitivity, or exposure, which could be useful for developing plans for reducing livelihood vulnerability to changing climate and related hazards, LVI identified the important component(s) and clustered sub-components that are the most significant drivers of vulnerability in the studied community [5]. Despite being strong, the vulnerability index development technique has certain drawbacks. One significant drawback of this strategy has been noted: the use of an equal-weighing scheme in LVI construction [29].

Despite the differences in their component values, the total LVI for each park was almost similar. The results suggested that the studied regions were moderately vulnerable. Similar findings were made in comparable places in Nepal [5]. In contrast, other studies [12,15,30,31] found a relatively lower LVI than our research. Additional sub-components such as energy, housing style, finance and income, land, infrastructure, and human–wildlife conflict may have influenced the outcomes. Literacy was one of the major subcomponents that had the most influence on the socio-demographic component. In Langtang NP, none of the family heads were literate, whereas, in Shey Phoksundo NP, just a handful had formal schooling. The fraction of female household heads that were educated was almost nonexistent. Many girls in rural and higher mountainous regions of Nepal are unable to advance to higher levels of education due to economic, social, cultural, and geographic obstacles [32]. Furthermore, poor literacy is caused by a lack of efficient education institutions in the higher Himalayan region. Compared to Shey Phoksundo NP, Langtang NP had fewer young people, and this may be attributed to its proximity to the capital city, where most of the young Nepalese migrate for work. Financial assets are easily convertible into other types of assets. As a result, they play a critical role in defining livelihood possibilities and adaptability to climate change [5]. Nomads in Shey Phoksundo NP appear to be more vulnerable in terms of finances and income than those in Langtang NP since they have less access to the banking system. Langtang National Park, the country's third largest revenue generator, has a greater tourism, trekking, and hotel industry, which may be the cause for improved banking facilities [33].

Health-wise, nomads in Shey Phoksundo NP were more susceptible than those in Langtang NP. The findings from Shey Phoksundo NP were significantly lower than those from [30,31], but they were still very similar to our findings from Langtang NP. Although the time it took to reach the nearest health post was twice as long in Shey Phoksundo NP, families in Langtang NP had a larger number of chronically ill individuals. All the herders surveyed in both national parks rely heavily on natural resources for food and energy. Even though they all raised their own limited seasonal food on the agricultural land and cooked mostly with forest firewood, due to the region's limited land base, poor soil fertility, cold climate, and short growing seasons, they had to acquire staple meals from the nearest market [34]. Other key reasons for the significant reliance on natural resources for food and energy might be the limited accessibility of road networks, which required nomads from both national parks to trek for several hours to reach the market or the nearest vehicle stop. Few households in both national parks were solely dependent on agriculture and livestock as a source of income. The variations in livestock and agricultural crops were very limited. Only two agricultural livelihood activities were undertaken in both national parks. In these regions, crop and livestock diversity may be severely constrained by harsh climatic conditions and rugged terrain [35].

Regarding climate change and disasters, more than two-thirds of the nomads in Langtang NP have suffered from climate change and disasters in the past five years. Although climate disasters caused higher crop, animal, and human losses in Langtang NP than in Shey Phoksundo NP, both national parks were less vulnerable to climate change and natural disasters. Our results were the same as those of one study [31] but did not agree with others [5,12,15,30]. Climate change and the disasters it causes have an influence on herders and indigenous communities, whose livelihoods depend primarily on natural resources [12,36]. The conflict between the imperative need for conserving wildlife and nature and the practical necessity of existence and a livelihood is an unavoidable and everlasting phenomenon. Both national parks appeared to have severe concerns with human–wildlife conflict, yet they were both somewhat vulnerable to the livelihood of the herders. The land-use practices of people living within the national parks, insufficient agricultural techniques, such as little or no fencing and inadequate livestock shelters, and allowing cattle to graze freely on open pastureland or wander into forests where predators naturally live can be possible reasons for the accelerated conflicts [37].

The analysis of LVI–IPCC showed that both national parks are moderately vulnerable. Similar findings were reported by other studies [5,12,30]. The exposure and sensitivity of Langtang NP are higher than those of Shey Phoksundo NP, but the adaptive capacity of Shey Phoksundo NP is higher than that of Langtang NP. Both the national parks are typically marked by high altitude and steep or rough topography, along with constant rain during the monsoon and extremely low temperatures with continual snowfall during the winter months. However, the slight difference in climate change-induced disasters and higher vulnerability to human–wildlife conflict might have made Langtang NP more exposed and hence more susceptible. Likewise, the lesser scope of crop diversification, as well as poor water, sanitation, and housing, might have made Langtang NP have a comparatively higher sensitivity. Additionally, poor market connections and transportation issues may heighten and worsen the adaptability capacity of Langtang NP. Overall, although herders in Langtang NP seem to be affected more, herders in both national parks fall under the least vulnerable group in this study.

## 5. Conclusions

For the first time in Nepal, this study assessed the vulnerabilities of nomads and their way of life to environmental changes, including climate change, disasters, and human–wildlife conflicts. The study assessed that both national parks were moderately vulnerable to the indications of the impact of climate change. The application of thirteen major components to produce LVI and the three contributing elements of exposure, sensitivity, and adaptive capacity to obtain the LVI–IPCC vulnerability indices are key aspects of this research. Our findings demonstrate the close connections among the various aspects of livelihood vulnerability. Natural resources, energy, water, and sanitation are the primary issues that impact the LVI since herders in both national parks reside in secluded mountain regions. Additionally, because all these factors are directly impacted by climate change, they will be considerably more susceptible in the near future. The livelihood of herders has been influenced by the recent rise in the occurrence of climate-related disasters, including floods, landslides, droughts, wildfires, etc., caused by the impacts of climate change. They have witnessed impacts on both their agricultural production and livestock.

Since residents of both parks have more communication options available in every house, giving them better emergency communication options, such as flood early warning systems and climate-smart agriculture practices can be a pivotal alternative. Techniques such as the introduction of drought-tolerant crops, the creation of seeds resistant to disease and pests, and proper compensation schemes can benefit the herders in both national parks. Similarly, in terms of long-term action, issues with livelihood should be resolved by implementing policy measures designed to minimize the sensitivity of habitat conditions, boost society's resilience, introduce sustainable livelihood alternatives, and improve individual stability. Strong policies such as promoting eco-tourism and sustainable agriculture/livestock practices can provide new income sources for local communities and help them adapt to changing environmental conditions. A comprehensive policy framework that integrates disaster risk reduction and climate change adaptation measures, as well as reducing greenhouse gas emissions, can address the root causes of vulnerability. Protecting the rights and interests of indigenous communities through their participation in decision-making processes and access to resources and services can ensure their well-being.

**Author Contributions:** Conceptualization, S.W.W.; methodology, S.W.W. and R.B.K.; software, S.W.W. and N.S.; validation, S.W.W. and R.B.K.; formal analysis, S.W.W., R.B.K. and N.S.; investigation, R.B.K., N.S. and S.W.W.; resources, S.W.W. and W.-K.L.; data curation, R.B.K., N.S. and S.W.W.; writing—R.B.K., N.S. and S.W.W.; writing—review and editing, S.W.W. and N.S.; visualization, R.B.K., S.W.W. and N.S.; supervision, S.W.W. and W.-K.L.; Project administration, S.W.W.; funding acquisition, S.W.W. and W.-K.L. All authors have read and agreed to the published version of the manuscript.

**Funding:** This research was supported by the Core Research Institute Basic Science Research Program through the National Research Foundation of Korea (NRF), funded by the Ministry of Education (NRF-2021R1A6A1A10045235).

**Data Availability Statement:** The data presented in this study are available on request from the corresponding author. The data are not publicly available due to privacy.

**Acknowledgments:** The authors would like to thank the National Research Foundation of Korea and the OJEong Resilience Institute at Korea University for supporting this study. In addition, we would also like to thank the nomads for voluntarily participating in this study and the government of Nepal for allowing us to conduct this study.

**Conflicts of Interest:** The authors have no conflict of interest.

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
