# Peer review of "Assessing the Livelihood Vulnerability of Nomads to Changing Climate in the Third Pole Region of Nepal"

_land, doi:10.3390/land12051105_

Round 1

Reviewer 1 Report

This manuscript is unique in highlighting an area that has not received much attention regarding how climate change and related risks affect the livelihoods of these marginalized communities in the mountainous Himalayan region.

 And I believe that the manuscript is of a very high level. However, there are some issues, such as insufficient information on the survey sites and redundant descriptions in the results section.

I would like to ask the authors to revise the manuscript, including the following points

1. Introduction 82

I don't think the bracket part is necessary because they are obvious

1. Introduction 89-90

Why did you use LVI and LVI-IPCC. Please mention the reason why and how applicable this method

1. Introduction 93

The authors explained, “how gender and job affect .. But based on your results, you described not only gender and job but also other aspects.

2. Materials and Methods 121-122

It is difficult to understand the relationship between the local people interviewed and these NPs. Do they live inside these NPs, or do they live in the surrounding area, or outside of these NPs? And what about the natural condition of these NPs in the surrounding area where these local people live? Please show the general information of target areas.

2. Materials and Methods 125-128

Why is the number of samples collected in both villages largely different?

2. Materials and Methods 132-134

According to the method, the survey team tried translating questions into the local language as much as possible.

The author mentioned that there were some parts of the questionnaire in this survey that could not be translated into the local language. How was the interview survey conducted for those sections?

3. Results 234236

The authors stated that average age of 48.8% and 44.41% in Langtang NP and Shey Phoksundo NP I am wondering if this number is not a percentage, but rather age.

3. Results 238-239

In this manuscript, the role and position of the community are very important. However, it is for me to understand difficult the meaning of "community ". However, it is for me to understand difficult the meaning of "community". Does community mean traditional village? Or administrative unit, or other organization?

3. Results 243

What is the “community-based groups?

3. Results 245-246, 219

Regarding local people's access to information, the author mentioned television in one section, but what is the position of television for them? There is no particular statement in this section.

3. Results 254-256

I still do not understand what you mean by " rely on for income". The author states that local people rely on natural resources for all of their income, while in another section, the authors stated that agriculture and livestock income is limited. What natural resources for income other than agriculture and livestock were there?

3. Results 293-298

What specific crops or livestock were preyed upon by which wild animals? If you have this information, please provide a brief description.

4. Discussion 376-378

The content described in this sentence is related to the livelihood patterns of the local people living in the NP. These contents should be mentioned in the study site section.

4. Discussion 390-391

The contents described in this sentence are also related to the overview of the survey site. It would be good to explain these in the study site section as well.

Reviewer 2 Report

Climate change has become a human consensus. Based on first-hand survey data, the authors used the IPCC theoretical framework to analyze the livelihood vulnerability of nomads in Nepal's third pole region under climate change conditions. Overall, the research topic selection has some significance and the results analysis is relatively solid. However, there are some problems with the study design. Here are some suggestions for your reference:

       (1) The introduction needs a modest rewrite. One is the lack of clarity about the key scientific question the study is trying to address, which makes the whole study feel like a technical report rather than a rigorous academic paper. Second, compared with the existing studies, it seems unclear where the marginal contribution of this study is. In fact, as far as I know, there have been a large number of studies in the academic circle using the theoretical analysis framework used by the author in this paper to make corresponding analysis. It's not clear what makes this study unique compared to those studies. It's not clear. It's hard to get published.

       (2) Authors need to strengthen the dialogue with theoretical research. In the current version, some basic theoretical analysis framework and several core concepts under the theoretical analysis framework are not introduced correspondingly, which is difficult to accept. If these basic concepts are not explained, the selection of indicators will be particularly arbitrary and lack of corresponding basis.

       (3) Discussion and policy implications need to be further strengthened, especially the differences and possible causes from the existing large number of similar research results.

Minor editing of English language required

Reviewer 3 Report

This article is missing several parts that don't allow to get a decision on the validity of the analyses and overall the article. Firstly, the authors don't explain how the sub-components (illustrated in Table 2) have been extracted and which data are used. In addition, it is important to quantify the size of the interviewed cluster and which kind of data has been analyzed. After explaining these key elements, the article must be implemented in the results and discussion sections. 

Round 2

Reviewer 2 Report

I have no other comments, thank you.

Minor editing of English language required